# Freshwater lake to salt-water sea causing widespread hydrate dissociation in the Black Sea

Vincent Riboulot [1], Stephan Ker[1], Nabil Sultan[1], Yannick Thomas[1], Bruno Marsset[1], Carla Scalabrin[1], Livio Ruffine[1], Cédric Boulart[1] & Gabriel Ion[2]

Gas hydrates, a solid established by water and gas molecules, are widespread along the continental margins of the world. Their dynamics have mainly been regarded through the lens of temperature-pressure conditions. A fluctuation in one of these parameters may cause destabilization of gas hydrate-bearing sediments below the seafloor with implications in ocean acidification and eventually in global warming. Here we show throughout an example of the Black Sea, the world's most isolated sea, evidence that extensive gas hydrate dissociation may occur in the future due to recent salinity changes of the sea water. Recent and forthcoming salt diffusion within the sediment will destabilize gas hydrates by reducing the extension and thickness of their thermodynamic stability zone in a region covering at least 2800 square kilometers which focus seepages at the observed sites. We suspect this process to occur in other world regions (e.g., Caspian Sea, Sea of Marmara).

[1] IFREMER, REM-GM, BP70, 29280 Plouzané, France. [2] National Institute of Marine Geology and Geo-ecology, RO-024053 Bucharest, Romania. Correspondence and requests for materials should be addressed to V.R. (email: riboulot@ifremer.fr)

Gas hydrates (GH) are present worldwide both on land (permafrost) and in ocean sediments[1–3]. The global magnitude of free gas enclathrated in hydrate form is staggering and may surpass the total conventional gas reserve[4]. GH are suspected to be the trigger of extensive geological hazards leading to large mass wasting[2,5,6] with potential tsunami hazards[7–9]. Additionally, the ubiquitous presence of GH close to the seafloor on continental margins makes their destabilization, due to a change in pressure by sea level changing[5,10] or temperature by warming of global ocean current[11,12] and seasonal fluctuation of the bottom-water temperature[13], a global process potentially affecting all world ocean margins[14].

GH is also suspected to be a potential contributor to ocean acidification[15] and climate change[16–18]. Rapid climate disturbances and negative carbon-isotope excursions have been linked to dissociation of GH and global warming[19]. GH destabilization can release significant quantities of gas (mostly methane) into the ocean, thus may affect methane inputs into the atmosphere[16,17] and leading to spikes in atmospheric carbon levels[20,21]. But the role of gas hydrate in global change is likely to be overestimated[22]. It is not yet known what fraction of methane reaches the atmosphere and the possible influence it may have on the climate[23].

Indirect evidence of the presence of GH in marine sediment can be obtained through the Bottom-Simulating Reflector (BSR) detected in seismic data. The BSR represents the base of the Gas Hydrate Stability Zone (GHSZ) that appears as strong, negative-polarity, high-impedance seismic reflections caused by free gas at the base of the phase boundary[24,25]. BSRs are often semi-continuous, crosscut stratigraphy in seismic sections, and their position can also be inferred on the basis of aligned amplitude terminations[26]. The cartography of the BSR in the Black Sea[1,27,28] exhibits its widespread occurrence, indicative of extensive development of hydrate accumulations.

Like other basins worldwide, such as the Sea of Marmara[29], the Black Sea was totally disconnected from the world oceans during its recent geological history (the late Quaternary Period)[30–32]. This long-term isolation induced a drastic change in the Black Sea water salinity[33] and mechanically modified the sediment pore water salinity and therefore, the hydrate equilibrium conditions[34]. Prior to the reconnection of the Black Sea with the Mediterranean Sea at ~9000 calendar year before present (9 kyrs cal BP), the Black Sea evolved as a fresh to brackish water lake with a water characterized by a salt concentration of 2 psu (Practical Salinity Unit). During the reconnection, via the Bosphorus (Fig. 1), hydrological changes were drastic. Bottom water salinities increased up from a couple of psu to ~22 psu and remained stable for the last 2500 yrs of the reconnection[33]. Here we demonstrate that this salinity change and its diffusion into the sediment has been inferred as the driving force behind the dissociation of gas hydrates over a large area of the Black sea.

## Results

**Gas hydrate characterization**. In the Romanian sector of the Black Sea (Fig. 1), BSR observation and velocity analysis from conventional High Resolution (HR) and deep towed Very High Resolution (VHR) seismic profiles, acquired during the recent GHASS cruise (10.17600/15000500), provide irrefutable evidence of the presence of the GHSZ (Fig. 2). The recovery of GH from three long Calypso © cores at 800 m below sea level (mbsl), into the first 6 m below seafloor (mbsf) of sediment, provide in situ evidence of the presence of GH close to the seafloor and represents the first collected samples in the Romanian sector of the Black Sea (Supplementary Fig. 1).

The gas leading to the formation of GH and the shallowest BSR observed in the Black Sea is mainly biogenic[35]. Gas sampled

on the continental slope on either side of the Danube canyon contains >99% of microbial $CH_4$. The biogenic $CH_4$ is formed during the early diagenesis stage in the evolution of organic materials in sediments and $CH_4$ forms structure sI hydrate[34].

Pore water measurements, carried out on 8 sedimentary cores underline a gradual fall in salinity from 22 psu at the seafloor level to near 2 psu at around 25 mbsf (Supplementary Figs. 2 and 3). Salinity measurements at the DSDP hole 379A in the central plain of the Black Sea are close to 3 psu for the next 250 m of sediment with one peak of 7 psu at around 80[36,37] (Supplementary Fig. 3). Under these salinity conditions, the GH phase boundary is expected to change significantly with depth and evolve over time with the salt diffusion through sediments. The consequence of such a transient diffusion process is an evolution in time of the GHSZ leading to GH destabilization as the pore water will become enriched in salt. The effect of salinity variations on the GHSZ has received much less attention in literature which remains largely dominated by the effect of pressure and temperature variations[5,10–13]. The Black Sea can therefore be considered as a natural laboratory for acquiring improved knowledge of the impact of salinity variations on the GHSZ extent. The geophysical, sedimentological and geochemical data acquired during the GHASS cruise provide the required input to predict GHSZ evolution over time and space.

The GHSZ was simulated using the two-dimensional GH stability model developed by Sultan et al.[38], with the GH forming gases containing 99.6% $CH_4$, a seawater temperature of 8.9 °C, a geothermal gradient of 24.5 °C/km and a salinity profile derived from the measured chloride profile over 25 mbsf (Supplementary Fig. 3). The results of the simulation are broadly consistent with the observed depth of the current BSR (Fig. 2). For water depths shallower than 750 mbsl, the model-predicted BSR is in agreement with the observed BSR depths derived from seismic data, whereas at greater depths, the predictions slightly diverge from observations, with the observed BSR consistently deeper than the predicted one (Fig. 2). A thermal re-equilibrium process may explain this divergence. During the Last Glacial Maximum (LGM), seafloor temperature was lower than the present-day[39] and its propagation through sediment takes longer to reach deep BSRs. Such thermal transient effect is often ignored when calculating the response behavior of the hydrate reservoir[40] but it was demonstrated that the Black Sea is currently experiencing such a transient state[41].

**Evolution of GHSZ over time**. The dynamics of GHSZ in the Black Sea has mainly been regarded through the lens of temperature- pressure conditions, whereas its evolution due to salt diffusion within sediment deserves as much consideration due to the recent history of the sea. The results of numerical calculations carried out using the approach developed by Sultan et al.[38], underline that the hydrate deposit is currently undergoing dissociation and allow us to project its fate in the near future (Fig. 3). In addition, these results show the evolution with time (0.2, 0.5, 1, 5, 10, 20, 50, 100, 200 kyrs) of the uppermost and basal boundaries of the GHSZ (Supplementary Fig. 4). For the 5000 yr-period calculations, temperature and pressure conditions are considered constant[42] and only the salt diffusion may decompose hydrate and affect the GHSZ. The model considers water-depth shallower than 750 mbsl because, as previously stated, the thermal re-equilibrium is not yet fulfilled at a greater depth (Fig. 2).

The calculation of the GHSZ with time and as a function of salt diffusion indicates that the present shallowest GHSZ is at 15 mbsf and displays a tongue-like shape with a length of 400 m and a

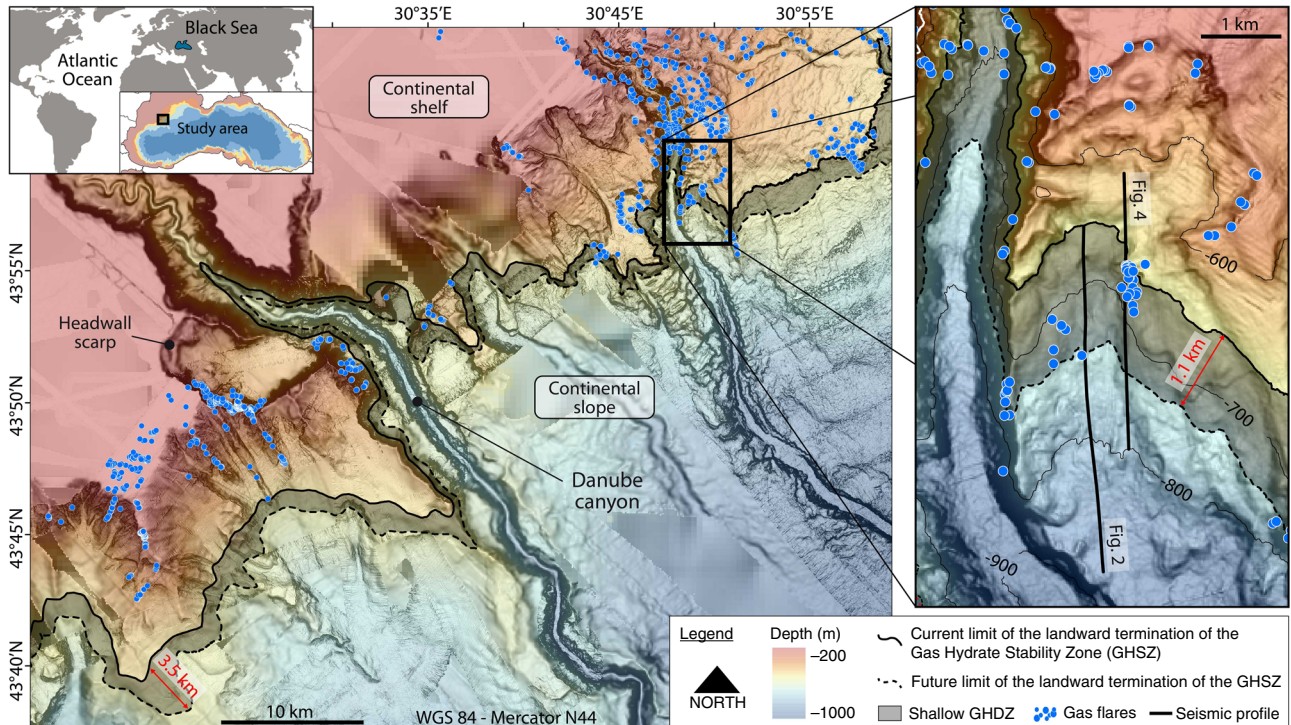

**Fig. 1** Gas Hydrate Destabilization Zone in the Romanian sector of the of the Black Sea margin. The GHDZ, area between the black line and the black dashed line, has been determined in the Romanian sector of the Black Sea where the continental shelf and slope are incised by the Danube canyon and several submarine scarps. The GHSZ is comprised between the modern and the future (5000 yrs) landward termination of the GHSZ that correspond to the 660 and 720 mbsl bathymetric contours respectively. Gas flares (blue points), non-randomly distributed, are located outside the GHSZ and within the GHDZ. They are observed within the shipborne RESON 7150 multibeam swath along the vessel track indicated in the Supplementary Fig. 2. The inset shows the study area with the location of the seismic profiles used in the present study

thickness of 20 m ranging between 680 and 710 mbsl (Figs. 2–4). The landward termination, the GH tongue-like shape, is predicted to decompose within the next 1000 yrs (Fig. 3). It will also evolve from 660 mbsl to 720 mbsl and draws a zone where GH becomes unstable and so called "Gas Hydrate Destabilization Zone" (GHDZ) (Figs. 1–4). Consequently, the diffusion of the salt through sediment affects the GHSZ and causes GH decomposition close to the seafloor. This salt related decomposition process probably contributes to the release of methane into the water column and may explain the present-day gas flares rising from the seafloor between 660 and 720 mbsl (Fig. 4).

Water column imaging, acquired with the multibeam echosounder (Supplementary Fig. 2), shows several huge gas flares in the study area mainly outside the GHSZ demonstrating the presence of large amounts of methane within sediment (Fig. 1). The gas flares can rise 400 m from the seafloor and terminate at around 50 mbsl. Within the GHSZ, upward gas migration from deep sources is expected to be obstructed by the hydrate low permeability or to be converted to solid hydrates thus preventing the occurrence of gas flares seaward 700 mbsl. Some gas flares are located at the landward termination of the GHSZ within the GHDZ. We attribute them to the current dissociation of GH due to salinization of sediment (Figs. 1 and 4). The gas flares observed at one site, inside the GHSZ and outside GHDZ, could be explained by the activity of a fault system (Supplementary Note 1, Supplementary Fig. 5).

As a first-order constraint on the GHDZ, we make a speculative extrapolation of our calculation results across the continental slope of the Black Sea taking into account the geology, the submarine sliding and its proven oil, free gas and gas hydrate potential (Fig. 5). We excluded the area close to the Bosphorus Strait because the salinity within sediment is not similar to the whole Black Sea[43]. Indeed, during the lacustrine stage where the

sediment deposits within fresh water conditions, this area was submitted to a very low sedimentation rate accompanied by a high erosion processes due to the hydrodynamic transients in comparison to the rest of the Black Sea[44]. Numerous GH samples and geophysical evidence of the presence of GH in the Black Sea show GHs are all along the continental slope of the Black Sea[36,45]. The Dnepr paleo-delta area west of the Crimea Peninsula, where the depth limit of 99.5% of the 2778 detected active seeps coincides with the phase boundary of GH, reveals that GHs formed at the base of the GHSZ act as an effective seal, preventing gas reaching the seafloor and the water column[46]. The dynamics of the GH in this area are similar to those observed in the Romanian sector of the Black Sea[47]. Also, free gas/GH systems discovered in the Georgian[48,49], Turkish[50,51], Bulgarian[27,52], Crimean[53], Russian[45] and Romanian[47,54] waters of the Black Sea demonstrated the occurrence of GH at the landward termination of the GHSZ at around 700 m water depth with a control on the spatial distribution of the gas plumes (Fig. 1 and Supplementary Fig. 2). This brief review suggests that an extrapolation of our results on the Romanian margin to an important part of the Black Sea margin is justified (see Supplementary Fig. 2 and references in caption). This approach reveals that the onset of methane hydrate will destabilize along the ~ 1700-km span of the Black Sea margin for centuries until the salinization of sediment is completed. We estimate that hydrates are currently destabilizing within a sedimentary volume of $4.3 \times 10^{10}$ m$^3$. Assuming an average porosity of 60% in shallow sediments where hydrates are destabilizing, hydrate filling 1–5% of the pore space of the sediment layer and the melting of 1 m$^3$ of hydrate may release about 165 m$^3$ of methane under standard temperature and pressure conditions[2,40], we estimate that ~ 4.2 × $10^{10}$ to ~ 2.1 × $10^{11}$ m$^3$ of methane are currently destabilizing

beneath a seafloor area of 2800 km² off the Black Sea margins (Fig. 5). Moreover, volume expansion from GH dissociation is believed to increase pore pressure in hydrate-bearing sediments along the slope edge, reducing slope stability in areas prone to frequent slope failure. If the expected ongoing hydrate destabilization triggers submarine landslides at this site, the amount of methane released could reach one order of magnitude greater[24,55].

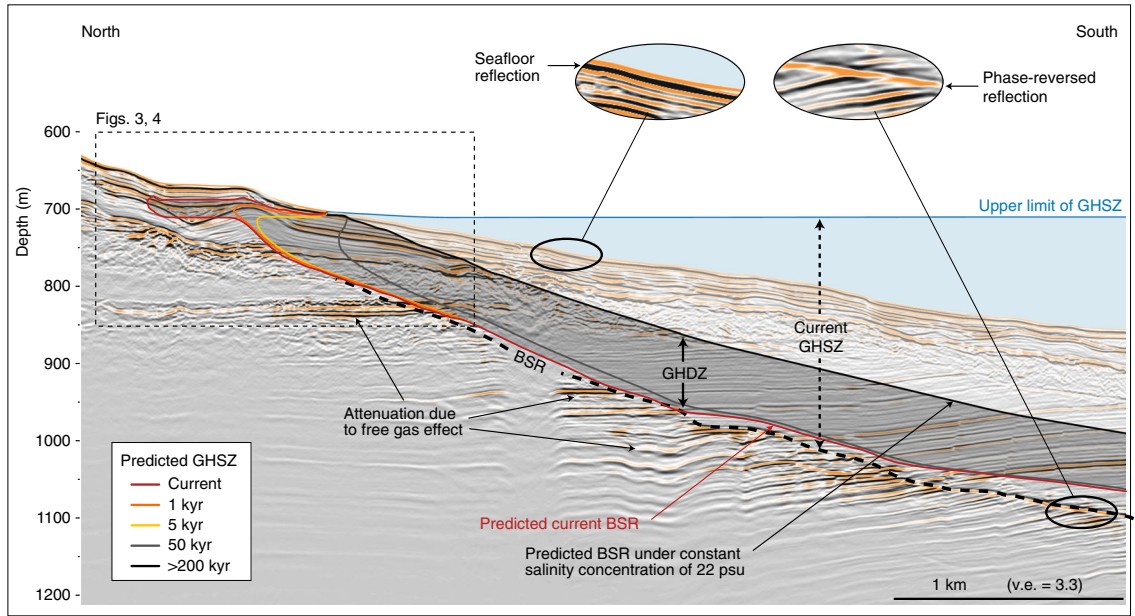

**Fig. 2** Current GHSZ and its predicted evolution over time under salinization of sediment. The multi-channel HR seismic line (post-stack depth migrated section) is acquired in the region showing a clear BSR indicative of the base of the GHSZ. The BSR is characterized by a strong and negative polarity reflector (black circles) that behaves erratically with depth beneath seafloor < 750 mbsl. The seismic profile is shown using the American convention where a decrease of impedance is represented by a negative reflection coefficient. To place first-order constraints on the timing of methane hydrate dissociation along this line, we run a time dependent diffusive salinity forward model. The resulting calculations show evolution of the methane hydrate stability zone over time. The base of the current GHSZ will evolve with time to the black line corresponding to the predicted GHSZ under a constant salinity concentration of 22 psu according to simulation. Each colored line represents the transient location of the model-predicted GHSZ in the future

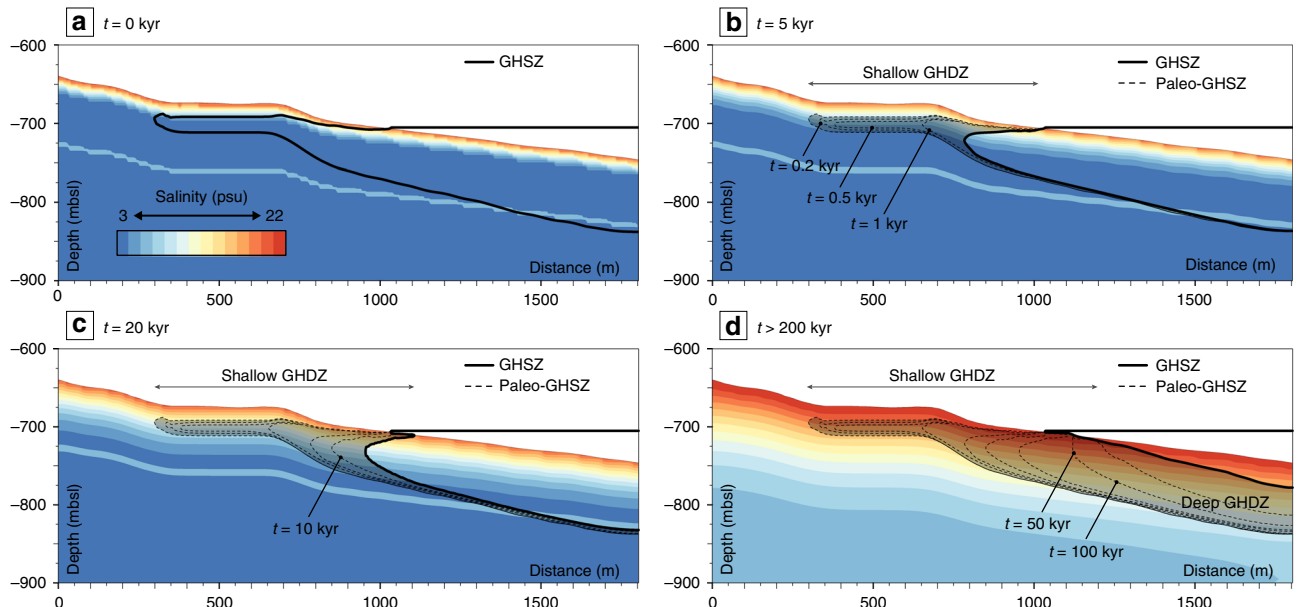

**Fig. 3** Simulation of the evolution of the GHSZ over time. The results of the simulation using the two-dimensional numerical model developed by Sultan et al.[38] are presented through 4 steps of the calculation: **a** t = 0 kyr, **b** t = 5 kyrs, **c** t = 20 kyrs, and **d** t > 200 kyrs. The landward termination of the GHSZ evolve over time. Hydrate deposit, the gray area, named "Gas Hydrate Destabilization Zone" (GHDZ) is currently undergoing dissociation caused by salt diffusion through sediments. This simulation is performed using the shallow part of the seismic profile presented in Fig. 2. All the steps of the calculation (t = 0, 0.2, 0.5, 1, 5, 10, 20, 50, 100 and >200 kyrs) are presented in the Supplementary Fig. 4

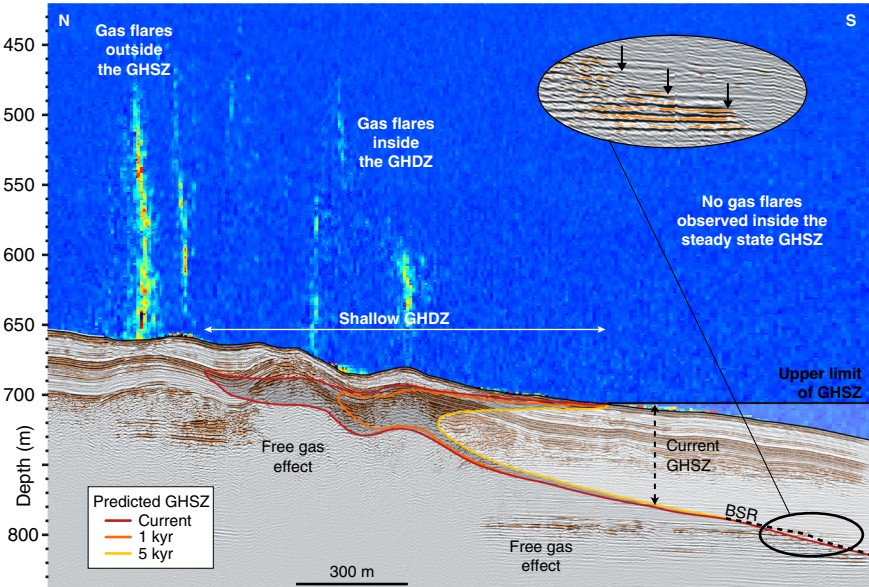

**Fig. 4** Methane hydrate dissociation due to salt diffusion responsible for gas flares. The interpretation of a pre-stack depth migrated VHR seismic section (acquired with the deep-towed Sysif multi-channel system) shows the free gas trapped under the BSR. It corresponds to an increase in the attenuation and amplitude anomalies (black arrows). Due to salinization of sediment, the dissociation of GH is expected to evolve over the next several thousand years (colored lines simulate the model-predicted evolution of the GHSZ at two different time steps). The processed water column echogram along the Sysif line (close-up view of part of the HR seismic profile presented in Fig. 2) shows the location of the gas flares outside the current GHSZ and inside the shallow GHDZ, which matches positively with the free gas location and the predicted GHSZ evolutions

The changing salinity of the Black sea, over the last 9000 years, has the potential to thaw and convert solid methane hydrates trapped below the seafloor into excess pore free gas/water mixture, increasing the potential for both slope failures[2,5,6] and tsunami generation[7–9]. The future evolution of the GHSZ provides a powerful mechanism for contemporary methane hydrate decomposition and may have a significant impact on margin shaping and the ocean carbon budget. Moreover, the presence in the Black Sea of potential unstable GH zones under the first 15 m of sediment is an important hazard source for the development of subsea structures and therefore economic activities. It is unlikely that the Black Sea margin is the only area experiencing widespread hydrate destabilization due to changing salinity within sediment. Recent studies have suggested that similar main driving mechanisms could occur in the Caspian Sea[56] and the Sea of Marmara[29,57].

## Methods

**Bathymetry and water column acoustic data.** Ship-borne multibeam surveys were conducted to map the continental external shelf and slope adjacent to the Danube canyon and to detect and locate the presence of free gas in the water column (Supplementary Fig. 2). This was performed onboard the R/V Pourquoi pas? during the 2015 GHASS expedition (doi:10.17600/15000500). The acoustic data were acquired with (1) a Reson seabat 7111 multibeam echo sounder for shallow water from 5 to 500 m (100 kHz, 301 beams, 1.8×1.5° beam width, 0.17–3 ms pulse length, up to 20 pings per second), and (2) a Reson seabat 7150 for mid and deep water from 200 to 2000 m (24 kHz, 880 beams, 0.5×0.5° beam width, 2–15 ms pulse length, up to 15 pings per second). The shelf and upper slope were surveyed by both sounders, while the deepest area was only surveyed by the Reson seabat 7150. Bathymetric resolution of the whole study area is 20 m. Water column processing was performed onboard with SonarScope and GLOBE softwares (© Ifremer).

**Seismic data.** The 2D High Resolution seismic data were recorded using a 96-trace streamer of 6.25 m group intervals, state-of-the-art Solid© technology. The single min-GI© air gun source, 24 cubic inches, provided a sharp seismic signature. The shallow immersion of both streamer and source gives rise to a 70–225 Hz frequency bandwidth signal. Seismic resolutions are in the order of 2.5 m vertically and 10–25 m horizontally. The achieved horizontal resolution depends on the quality of the imaging processing and on 3D side effects. The depth penetration within the

sediment ranges from 400 to >750 mbsf. For profiles with >500 m of water depth, penetration is limited by attenuation due to shallow gas and by the seafloor multiple. The relatively short source-to-receiver offset is well tailored to geophysical characterization of sediments up to around 600 m below the sea surface. For greater depth, we are able to infer general trends with more incertitude on the absolute velocity of propagation. After accurate positioning of shot points and receivers, data were sorted using a 6.25 m distance between Common Mid Points (CMP). Detailed velocity analyses were performed on super-gathers every 150 m: 5 combined CMP every 24 CMPs. The picking of RMS velocities was performed on semblance panels using a horizon consistent scheme. Interval velocities were computed using Dix's law. Post-stack Kirchhoff depth migration was then performed with Green functions derived from an Eikonal implicit solver using a smooth interval velocity model. Despite the relatively deep depth of the BSR, 900–1200 m, results are consistent. Where seismic sections display strong amplitude BSR, there is always an inversion of the velocity gradient related to gas-bearing sediments below the BSR. Moreover, the sediments above the BSR (layer of 100–200 m thickness) show high interval velocity (1800–2000 m/s) compared to sub-surface sediments (around 1600 m/s or less), which may be related to gas hydrate-bearing sediments.

The 2D very high resolution seismic data were acquired with the Sysif system[58,59]. SYSIF is a deep-towed seismic device hosting a piezoelectric transducer (seismic source) and a 120 m long seismic streamer (52 traces, 2 m). During the GHASS cruise, seismic profiles were acquired using a linear frequency modulated (100 ms, 220–1150 Hz) acoustic signal thus providing a horizontal resolution of 3 m and an infra-metric vertical resolution. The horizontal resolution is obtained after seismic processing where the imaging process improves the lateral resolution from the width of the first Fresnel zone down to the mean signal wave length.

**Gas composition.** We determined the nature of gas from the collected hydrate samples. The GH were allowed to decompose and the released gases were analyzed by gas chromatography. A gas chromatograph µGC R3000 from SRA equipped with a µTCD and a PoraPlot U capillary column was used. Methane was overwhelmingly present at more than 99.6%-mol. Such a high methane content for the hydrate sample implies a gas source with at least as much methane. Analytical precision was better than 3%.

**Salinity and temperature measurements.** Salinity profile was derived from the chlorinity profile (Supplementary Fig. 3a). Chloride concentration was measured by ionic chromatography (Dionex ICS 5000 EG from Thermo Scientific) from pore water samples collected along the calypso core with a depth resolution of 20–30 cm. Measurements were provided with an analytical error of > 2%.

During the GHASS cruise, the geothermal gradient was measured using temperature sensors fixed at specific intervals along a 12 m pipe. Seven probes were

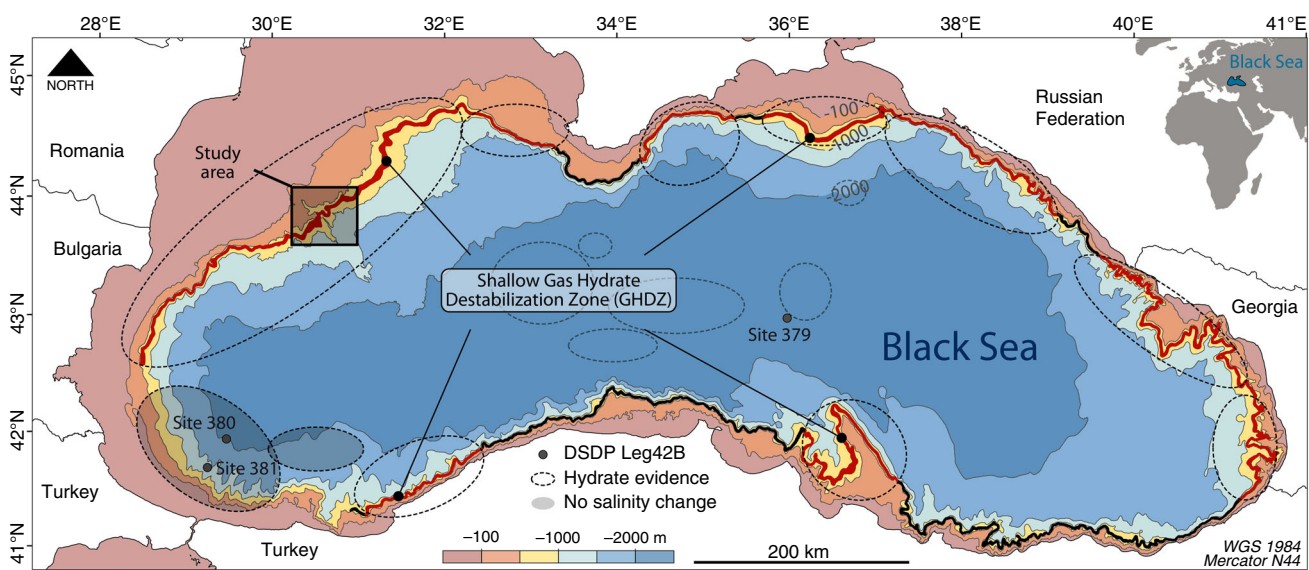

**Fig. 5** Gas Hydrate Destabilization Zone potentially covering 2800 square kilometers of the Black Sea margin. The Gas Hydrate Destabilization Zone (GHDZ-the red zone) close to the seafloor denotes the area where the GH has been recovered or inferred (see Supplementary Fig. 2 and references in caption) and are expected to decompose due to salinity diffusion through the sediment. The red zone is comprised between the modern and the future (5000 yrs) landward termination of the GHSZ that correspond to the 660 and 720 mbsl bathymetric contours respectively

fixed on supports welded helically around the pipe of the coring system. The thermal probes thus formed temperature sensors are fixed on supports which hold them at 7 cm from the wall of the pipe in order to avoid the measurement artifact related to the temperature disturbance introduced by the presence of the pipe within the sediment. The geothermal gradient measurement at the site CSF04 is ~ 24.5 °C/km (Supplementary Fig. 3b). The error range of measurements is 0.01°.

The CTD-rosette package consisted in a SeaBird 911+ fitted with 24 NISKIN bottles and various sensors for Conductivity, Temperature and Pressure, as well as two Seapoint Turbidity Meters (STM), which allow for the detection of high concentration of suspended particulate material. Real-time data from the CTD and Turbidity sensors were transmitted to the Seabird 11+ Deck Unit and computed using the SeaSave software with the usual data correction recommended by Seabird. All data were also averaged and filtered to discard the outliers, especially for turbidity. Vertical profiles were conducted throughout the water column, at speeds between 0.7 and 1 m/s, stopping ~ 5 m above the seafloor, in a single trip. Temperature was measured using the SBE 3+ T sensor, which contains a high-speed, pressure-protected thermistor working between −1 and +31 °C. Salinity was determined from the conductivity measurements performed by the SBE 4 C sensor. Units are given in PSU (Supplementary Fig. 3c).

**Numerical model of GHSZ**. Our working hypothesis concerning GH dynamics and the evolution of the GHSZ as a function of salt diffusion has been tested using the two-dimensional numerical model developed by Sultan et al.[38] The temperature and pressure of hydrate-phase equilibrium are calculated at each time step using the equality of chemical potentials in each phase at equilibrium as proposed by Van der Waals and Platteeuw[60]. Gas is transported in a dissolved phase, and is taken into account following Fick's law. The numerical model used fully accounts for the latent heat effects which may impede gas hydrate dissociation (self-preservation phenomenon). Indeed, additional heat source and heat sink are produced as gas hydrate forms and dissociates respectively[61]. Gas hydrate dissociation is also known to cause a local decrease in salinity which may also impede the decomposition process. However, for the considered timescale calculations (kyrs), the small salt perturbation occurring at the hydrate border is expected to be second order with respect to the general process generated by the vertical salt diffusion.

At each calculation step, the fluid flow (Darcy's law) and the molecular diffusion of salt ions (Fick's law) within the pore water of sediment are simulated at each node from initial and boundary conditions by using the finite difference method. The calculation domain was divided into 10 different layers. The results of the simulation are presented through 10 steps of the calculation ($t = 0, 0.2, 0.5, 1, 5, 10, 20, 50, 100,$ and $>200$ kyrs) (Supplementary Fig. 4). They underline that the hydrate deposit is currently undergoing dissociation and allow us to project its fate in the near future.

The thermal boundary conditions are zero thermal flux on the two vertical extremities of the calculation region. A seafloor temperature of 8.85 °C and a thermal gradient of 24.5 °C/km at the base of the calculation domain were considered constant for the entire calculation period (Supplementary Note 2,

Supplementary Figs. 3 and 6). The salinity boundary conditions are zero ion flux on the two vertical extremities and the base of the calculation region with a salt concentration of 22 psu at the seafloor. The initial salinity profile used was equivalent to the one presented in Supplementary Fig. 3a and the chloride diffusion was calculated using a mean diffusion coefficient for chloride ions including tortuosity equal to $6 \times 10^{-10}$ m$^2$/s.

**The salt diffusion coefficient used for the simulation**. The chloride diffusion coefficient defined in Schulz and Zabel[62] is $1.32 \times 10^{-9}$ cm$^2$ s$^{-1}$. We have integrate the tortuosity defined in Boudreau et al.[63] and have obtained $6 \times 10^{-10}$ cm$^2$ s$^{-1}$ for the chloride diffusion coefficient (Supplementary Note 2, Supplementary Fig. 7). This coefficient is used for the calculation of the predicted GHSZ and the simulation of its evolution over time.

**Data availability**. Data sets used in the current study are acquired during the GHASS expedition (10.17600/15000500) and data set information are available in the GHASS cruise report http://archimer.ifremer.fr/doc/00300/41141/. All data are available within the Article and Supplementary Files, or available from the authors upon reasonable request.

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

## Acknowledgements

The support by officers and crew during the GHASS cruise on board R/V Pourquoi Pas? (2015) is greatly appreciated, as is the dedication of the Genavir and Ifremer technical staff during the cruise. We thank sincerely Alison Chalm for her revision of the English language. We thank C. Berndt and M. Haeckel for providing insightful comments.

## Author contributions

V.R. devised the approach, wrote the paper, performed the figures, contributed to the simulation and was co-chief of the GHASS cruise. S.K. conceived and supervised the Black Sea project at IFREMER, contributed to the editing of the paper and the approach,

contributed to the SYSIF acquisition, processing and interpretation, was co-chief of the GHASS cruise. N.S. instigated the Black Sea project, elaborated and performed the simulation and contributed to the editing of the paper. Y.T. acquired, processed and interpreted the HR seismic data. B.M. developed the SYSIF, acquired, processed and interpreted the SYSIF seismic data. C.S. acquired, processed and interpreted multibeam data and contributed to Fig. 4. L.R. collected sediment samples and analyzed gas composition and salinity concentration within pore water. C.B. acquired, processed and interpreted water column salinity and temperature with the CTD rosette. G.I. collaborate to the project and the interpretations, participate to the GHASS cruise and to obtain authorization to acquired data in the Romanian sector of the Black Sea.

## Additional information

**Competing interests:** The authors declare no competing financial interests.

