## [Peer Review File · Nature Communications]

Reviewers' comments:

Reviewer #1 (Remarks to the Author):

Natural gas hydrates are considered as future potential energy source. As well as gas production studies from these reservoirs, the environmental effect of gas hydrate dissociation should be investigated. This study investigates especially the Black Sea gas hydrate dissociation with the effect of salt diffusion. It is a well-organized study. I advise the acceptance of this manuscript after the following minor revisions are done:

This study summarizes:

- 1) Abstract should be revised because it is hard to understand what it is exactly done in this study with current abstract.
- 2) Line 73 "...the GH forming gases containing 99.6 % CH₄, a seawater..." what is the composition of other 0.4 % gas?
- 3) Line 116-117, "...Assuming hydrates occupy 1% to 5% of the sediment layer, and..." is there any specific reason of the selection of these hydrate saturations?

Reviewer #2 (Remarks to the Author):

This paper discusses gas hydrate destabilization in the Black Sea. The authors posit that the known methane hydrate reservoirs in the seafloor sediment there are dissociating, and will continue to do so, as a result of increasing pore water salinity. They proceed to test this hypothesis by comparing the results of an equilibrium model with seismic (BSR) and other data.

I found the paper to be interesting and the results plausible. It is reasonably well-written and I had no problem with the grammar or logical progression. Having worked on the effects of ionic salts on hydrate stability in multi-component systems, the inhibiting influence of salinity is obvious and the unique, natural systems presented by the Black Sea with respect to a slowly-evolving solvent (i.e., liquid phase) field provides a good opportunity for in situ observation of salt-driven hydrate decomposition. It is refreshing to read a paper that views the stability of hydrate reservoirs through a more comprehensive thermodynamic lens that considers concentrations as well as merely temperature and pressure.

The field data appear to be of good quality, but seismics is not my area of expertise. Since the modeling results are a key component in testing the hypothesis, however, I feel that the manuscript would benefit from a few minor additions to address some omissions and make it self-contained (i.e., complete). Specifically, in the Methods section on the topic of Numerical Model of GHSZ (lines 200 to 217), please include a few comments on how the endothermicity of decomposition and associated release of fresh water--which locally affects temperature and salinity, respectively, in this slow diffusive system--is handled. Perhaps these effects were found to be trivial relative to the salinity influx and thermal gradient over the timescales considered. If so, then please comment accordingly, since such self preservation phenomena are of interest to the readership.

In addition to the above, I suggest some small changes in wording to avoid the appearance of hyperbole. Notably, in Lines 122-123: "...the amount of methane released could reach one order of magnitude greater." This is rather quantitative. How was this estimated? If there are no calculations to justify the increase by an order of magnitude, then it should be re-worded. Also in Lines 126-128, the authors speculate about possible impacts related to tsunami generation and changes to the ocean carbon budget. Does the bathymetry suggest that a slump has a reasonable chance of generating a significant tsunami effect? Also, while the carbon budget in the Black Sea

appears to be currently affected by methane release from hydrates, this is a local phenomenon that may or may not change significantly over time. I doubt that salt-driven hydrate destabilization will impact the global carbon budget.

Aside from the additional information about the model and some minor wordsmithing, I am favorably impressed by this manuscript.

Reviewer #3 (Remarks to the Author):

General comments

The paper claims that the effect of diffusion of salt from the seabed to the BSR can trigger dissociation of gas hydrates in basins characterized by recent "salinity inversion". The idea is novel and it is probably the first time that such workflow is applied in this specific context. The same concepts are potentially applicable to other basins worldwide.

The claims are not however convincing because of the lack of data to constrain the salinity at the BSR. The salinity profile is indeed limited to the first meters of sediments and far away from the BSR. This preclude to understand whether the GH are in a stable conditions. The authors should provide evidence of this. If the salinity at the BSR is close to the one of the seabed the GH are stable and will not dissociate in the near future.

The salt diffusion model proposed is convincing. In this case, the authors should however describe what the limitations of the model are and explain the approximations applied. How can we approximate with one single value for water temperature and a geothermal gradient based on the first 12m of sediments below the seabed the evolution of a basin in a time interval of 5k years? And the effects of erosion? Sedimentation rate? It is important to mention that most of the data use here is potentially linked to the presence of submarine canyons.

The extrapolation of the observation from the study area to the entire Black Sea is not convincing. The authors not only assume a nearly homogenous distribution of GH but also that the Temperature-Pressure-Salinity conditions are homogenous. This make the first order constrain of the GH destabilization zone very weak.

I do encourage the authors to check carefully the literature about the other basins before mention them as potential analog of the Black Sea.

I suggest the authors to provide the appropriate information in order to validate the method or to consider rewriting the manuscript in an area where the key information are available.

The manuscript is well-written and reads clearly. There are no ethical concerns present. The figures are clear and well edited.

Detailed comments

Lines 9-18 (Abstract): I suggest revisiting the abstract.

Lines 19-31: the paragraph is well-written but contains too much information on why gas hydrates (GH)

are important and less on the general aspects of GH. I suggest rewriting this paragraph and trying to get

the reader ready to get the rest of the manuscript.

Lines 32-35: the BSR, as discussed by Xu and Ruppel 1999, does not coincide to neither the base of the

gas hydrate stability zone (GHSZ) nor the top of the free gas zone (FGZ). I suggest referring to this paper

for a better significance of the BSR. I would explain also what a negative polarity is. It is negative with

respect to what. If the polarity of the seismic data is European the BSR is positive.

Lines 36-38: provide a reliable distribution map of the BSR, which you will use then to infer the distribution of the GH. Without the distribution of the GH in the Black Sea, most of your argument from

line 112 onwards are hard to argue (even for a first order evaluation). Reference 1 (fig. 4) shows a map

extracted from a work of Vassiles et al 2006 showing evidence of extremely variable GH distribution in

the Black Sea; References 26 and 27 show the distribution of GH only in NW Black Sea (both works are

based on 2D seismic data only). Reference 26 (fig.1) shows indeed just some areas covered by BSR.

Line 39: Reference 28 shows evidence of freshening consistent with a dehydration caused by smectite to

illite process.

Lines 47-49: a map showing more information of the actual salinity of the Black Sea is needed. What is

the impact of the rivers on the salinity of the Black Sea? You cannot extrapolate your model on the entire Black Sea.

Lines 58-59: please specify if this gas is coming from dissociation of GH or free-phase gas.

Lines 63-63: the salinity is measure to a depth of 25m How you know that in the deeper section (e.g. at

the BSR) higher salinities are not present? Reference 29 (fig. 5 top) shows indeed an increase of salinity

with depth below 40m (The simulated dissolved Cl⁻ profile retains the memory of marine-like waters of

the last interglacial...). Provide a salinity profile for the entire GHSZ.

Lines 72-74: here you are using 1D geothermal gradient (based only on the first 12m of sediments!!!),

one seawater temperature (!) and 99% methane to run simulations which you then apply to the entire

Black Sea! Please explain all the limitations.

Lines 78-79: divergence of BSR predicted/observed are probably the result of a constant geothermal

gradient based on the first 12 m of sediments. Revisit this interpretation once the GHSZ is calculated

more precisely.

Lines 90-93: please be specific on the limitation of the method. Sedimentation rate? Effect of freshening

of rivers? Erosion? Lithology? Your study area and most of the derived data is nearby a submarine canyon. What does this imply?

Lines 94-111: to discuss again once all the points suggested above are taken into account.

Lines 122-130: I suggests revisiting this once you have provided enough information to calibrate your

model. As it is now this is not adequate even for a first order constrain.

Lines 131-133: provide adequate references, as some of the cited ones (Reference 28 and 41) do not

mention the effect of salinity on the destabilization of GH.

Subject: paper NCOMMS-16-26092 - Response to reviewers.

Reviewer 1:

1) Abstract should be revised because it is hard to understand what it is exactly done in this study with current abstract.

We have partly rewritten and reorganized the abstract taking into account this comment. See lines 11 to 18 of the revised manuscript.

2) Line 73 "...the GH forming gases containing 99.6 % CH₄, a seawater..." what is the composition of other 0.4 % gas?

The table below shows the composition of the gas sampled in the study area.

Hydrate-bound gases	N ₂	CH ₄	CO ₂	C ₂ H ₆	C ₃ H ₈
Molecular composition %-mol	0.3796	99.5553	0.056	0.0071	0.0002

3) Line 116-117, "...Assuming hydrates occupy 1% to 5% of the sediment layer, and..." is there any specific reason of the selection of these hydrate saturations?

The 5% are justified by the amount of hydrate recovered from the three cores sampled in the study area which was about 10% of the whole cores. However, and based on previous published data, we introduced in the revised manuscript a hydrate concentration between 1% and 5% of the porosity (60%). Details are added in lines 128 to 131 of the revised manuscript.

Reviewer 2:

1) In the Methods section on the topic of Numerical Model of GHSZ (lines 200 to 217), please include a few comments on how the endothermicity of decomposition and associated release of fresh water--which locally affects temperature and salinity, respectively, in this slow diffusive system--is handled. Perhaps these effects were found to be trivial relative to the salinity influx and thermal gradient over the timescales considered. If so, then please comment accordingly, since such self-preservation phenomena are of interest to the readership.

The numerical model used fully accounts for the latent heat effects which may impede gas hydrate dissociation (self-preservation phenomenon). Indeed, additional heat source and heat sink are produced as gas hydrate forms and dissociates respectively (Sultan et al., 2004). Gas hydrate dissociation is also known to cause a local decrease in salinity which may also impede the decomposition process. However, for the considered timescale calculations (kyrs), the small salt perturbation occurring at the hydrate border is expected to be second order with respect to the general process generated by the vertical salt diffusion. This part was added to the manuscript lines 217 to 223 of the revised manuscript.

2) In addition to the above, I suggest some small changes in wording to avoid the appearance of hyperbole. Notably, in Lines 122-123: "...the amount of methane released could reach one order of magnitude greater." This is rather quantitative. How was this estimated? If there are no calculations to justify the increase by an order of magnitude, then it should be re-worded.

It is true, we don't have a numerical estimation to justify our purpose although we consider that the hydrate decomposition and the subsequent slope instabilities may change the T/P/S limit conditions and therefore amplify the hydrate decomposition process (Maslin et al., 2010). We agree with the reviewer that this sentence is more speculation than demonstration and if necessary can be removed from the text.

3) Also in Lines 126-128, the authors speculate about possible impacts related to tsunami generation and changes to the ocean carbon budget. Does the bathymetry suggest that a slump has a reasonable chance of generating a significant tsunami effect?

Studies on recent and historical tsunamis recorded in the Black Sea show that their most frequent cause was seismic activity but cases of tsunamis triggered by landslides were also described (Papadopoulos et al., 2011; Rangelov et al., 2008). The generation of tsunamis from submarine landslides or slumps depends mainly on the slump volume that seems to control both maximum wave height and maximum length of affected coastline (Papadopoulos and Kortekaas, 2003). Modeling results of tsunami generation carried out by Schnyder et al. (2016) show that a landslide occurring in 500/600 m water depth on a regional slope of 3°, similar to the present study area, can generate a tsunami. In the present study area, the headwall scarps with a missing sediment volume of 0.5 km³ is comparable to the landslide volume used by Schnyder et al. (2016) to simulate tsunami generation. We have added the reference "Schnyder et al., 2016" to the manuscript.

4) Also, while the carbon budget in the Black Sea appears to be currently affected by methane release from hydrates, this is a local phenomenon that may or may not change significantly over time. I doubt that salt-driven hydrate destabilization will impact the global carbon budget.

The basin-wide fluxes of methane of the Black Sea contribute to the global CH₄ budget (Kessler et al., 2006). Although the contribution of decomposing clathrates to the global CH₄ budget remains a major uncertainty (Reeburgh et al., 2003), we expect the salt-hydrate destabilization to contribute to the basin-wide fluxes of methane of the Black Sea that, in turn, may contribute to the global CH₄ budget.

Reviewer 3:

1) The claims are not however convincing because of the lack of data to constrain the salinity at the BSR. The salinity profile is indeed limited to the first meters of sediments and far away from the BSR. This preclude to understand whether the GH are in a stable conditions. The authors should provide evidence of this. If the salinity at the BSR is close to the one of the seabed the GH are stable and will not dissociate in the near future.

Sediment pore water salinity was acquired during the 379 DSDP project (hole 379A – Calvert and Batchelor, 1978) and salinity data are available from the first 600 mbsf. The calculation carried out in the present work considered the first 250 m of sediment while the sediment thickness affected by the salinization of the sediment during the next 5000 years period concerns only the first 50 m. The results of the simulation shown in the previous version of the paper remain unchanged by introducing in the calculation the real pore water salinity shown below where the salinity increases slightly at round 80 mbsf.

Distribution of the chlorinity of pore water at the DSDP hole 379A in the central plain of the Black Sea (modified from Meray and Sinayuc, 2016 and Calvert and Batchelor, 1978). This figure was integrated to the Supplementary Fig 2.

2) The salt diffusion model proposed is convincing. In this case, the authors should however describe what the limitations of the model are and explain the approximations applied. How can we approximate with one single value for water temperature and a geothermal gradient based on the first 12m of sediments below the seabed the evolution of a basin in a time interval of 5k years? And the effects of erosion? Sedimentation rate? It is important to mention that most of the data use here is potentially linked to the presence of submarine canyons.

The calculation of the mean geothermal gradient using the depth of the current BSR in the Black Sea (Zander et al., 2017) is in agreement with the in situ measurements presented in the present study. Uncertainties related to geothermal gradient are also analyzed through parametric studies considering thermal gradients between 21 and 28 °/km. These parametric studies and the consequence on salt-hydrate destabilization processes are shown in Supplementary Figure 04.

Concerning sedimentation versus erosion on the slope, Constantinescu et al. (2015) show that the sediment balance for the last 8 kyrs was close to 0 cm/kyr. Therefore, for the 5 kyrs calculation period, we considered unchanged bathymetry and constant temperature and pressure conditions (Past Interglacials Working Group of PAGEs, 2016).

3) The extrapolation of the observation from the study area to the entire Black Sea is not convincing. The authors not only assume a nearly homogenous distribution of GH but also that the Temperature-Pressure-Salinity conditions are homogenous. This make the first order constrain of the GH destabilization zone very weak.

We have rewritten this section of the manuscript and based on data reported from literatures we've explained why, as a first approximation, this simple extrapolation is justified. See line 114 to line 126 of the revised manuscript.

4) I do encourage the authors to check carefully the literature about the other basins before mention them as potential analog of the Black Sea.

This has been done and the three others basins mentioned in the study may undergo the same dissociation process due to salt diffusion. We have added some references showing the presence of hydrates at the landward termination of the GHSZ and the salinity gradient of these three other areas.

It may be noted that we have enriched the manuscript with 17 references.

Reviewers' comments:

Reviewer #2 (Remarks to the Author):

After reviewing the revised manuscript and the authors' rebuttal to my earlier comments, I am satisfied with their responses and believe that the manuscript warrants publication

Reviewer #3 (Remarks to the Author):

The paper claims that diffusion of salt in pore space in the first meters of sediments triggered by changes in salinity of the seawater can destabilise the gas hydrates with a subsequent release of methane into the atmosphere.

The idea is novel and of great interest however the extrapolation of the idea to the entire black sea is still not convincing. I suggest to review the amount of gas release from the gas hydrate destabilisation zone once a clear distribution of gas hydrates and salinity is taken into account.

I prepared a document (attached) with comments not addressed in the previous revision and new comments based on this version of the manuscript.

Subject: paper NCOMMS-16-26092B - Response to reviewers.

Reviewer 2: There is no comment.

Reviewer 3: **Response to comments not addressed from the previous review**

Lines 32-35: the BSR, as discussed by Xu and Ruppel 1999, does not coincide to neither the base of the gas hydrate stability zone (GHSZ) nor the top of the free gas zone (FGZ). I suggest referring to this paper for a better significance of the BSR. I would explain also what a negative polarity is. It is negative with respect to what. If the polarity of the seismic data is European the BSR is positive.

The BSR, as discussed by Xu and Ruppel 1999 differs from the BSR observed in our study area because velocity analysis of the seismic data shows a sharp decrease in the velocity right under the BSR revealing the presence of free gas trapped under the BSR. Conversely an increase in the velocity right above the BSR suggests the occurrence of hydrates at the base of the GHSZ corresponding to the location of the BSR.

The inverse polarity of the BSR is with respect to the polarity of the seafloor. So if the seafloor is positive, the BSR is negative.

Lines 36-38: provide a reliable distribution map of the BSR, which you will use then to infer the distribution of the GH. Without the distribution of the GH in the Black Sea, most of your argument from line 112 onwards are hard to argue (even for a first order evaluation). Reference 1 (fig. 4) shows a map extracted from a work of Vassilev et al 2006 showing evidence of extremely variable GH distribution in the Black Sea; References 26 and 27 show the distribution of GH only in NW Black Sea (both works are based on 2D seismic data only). Reference 26 (fig.1) shows indeed just some areas covered by BSR.

The Reference 1 (Merey and Sinayuc, 2016) uses a map from Vassilev et al. (2006) showing in blue the extension of the first BSR covering almost the whole Black Sea; the other colours represent the occurrence of multiple BSRs. In the study of Merey and Sinayuc (2016), the map presented in Fig. 6 shows the volume of CH₄ in hydrates in standard conditions for the Black Sea suggesting the occurrence of hydrates in our region of interest where hydrate may dissociate by salt diffusion.

The Reference 26 (Popescu et al., 2006) shows areas covered by BSR in NW Black Sea where 2D seismic data were available. The extended data acquired in the GHASS project in 2015, some of which are shown in our study, show that the BSR is observed in all seismic data located to the north of the Danube canyon at a water depth in agreement with the base of the gas hydrate stability zone. The cartography of this BSR is not the aim of the present paper but it may be added to the supplementary data if you consider it as essential.

Line 39: Reference 28 shows evidence of freshening consistent with a dehydration caused by smectite to illite process.

Indeed the authors of this paper show that the source of fluid freshening in the central part of the Ulleung Basin (Japan Sea or East Sea) is due to dehydration caused by smectite to illite process. This process was not described in the Black Sea.

Lines 47-49: a map showing more information of the actual salinity of the Black Sea is needed. What is the impact of the rivers on the salinity of the Black Sea? You cannot extrapolate your model on the entire Black Sea.

Water exchange is low between the Black Sea and the Mediterranean Sea due to the narrow Bosphorus Strait. Fresh water entering the Black Sea by river discharge is less dense than water from the Black Sea, mixing of fresh water/sea water takes place in the upper part of the water column, above 90 meters. Under the pycnocline (the water density limit that restricts vertical mixing and exchange between the deep layers and the mixed layers), the water column of the Black Sea is completely anoxic. The salinity

measurements in the water column of the Black Sea is close to 22 psu above 300 m water depth (e.g. Tuğrul et al., 2014; Spencer and Brewer, 1971). So the Black Sea salinity is close to 22 psu at the seafloor under 300 m water depth.

Reviewer 3: Response to comments on the revised manuscript

Line 45-49: as already commented on the previous version of this manuscript a detailed map of the seabed salinity is necessary before to calculate the amount of CH₄ released from the destabilization of the gas hydrates. Although the authors provide evidence of low salinity at site DSDP 379, at DSDP 380 and 381 the salinity is very different (e.g. from Manheim, 1978): These values show that the salinity is not so low below the seabed and that the gas hydrate are stable!

For the map of the salinity, please refer to the previous comment (Black Sea salinity is close to 22 psu at the seafloor under 300 m water depth).

Concerning the values of salinity at DSDP 380, 381, it is true that the value close to the Bosphorus Strait is not similar to the whole Black Sea. Indeed, during the lacustrine stage where the sediment deposits within fresh water conditions, the areas of the 2 DSDP 380 and 381 were submitted to a very low sedimentation rate in comparison to our study area and the rest of the Black Sea (Gillet, 2004). This low sedimentation is due to the lack of rivers and is accompanied by a high erosion processes due to the hydrodynamic transients that characterize the area. We have now removed this area from the extrapolation calculation.

Line 54-57: Gas hydrates have been cored at a depth of 6m. The depth of the core does not match with the region of dissociation of the gas hydrate. Looking at figure 5 the top of the tongue-shaped gas hydrate stability zone is indeed located far below the cored interval and at a depth of 20-30 m below the seabed. It would be important to understand the impact of the dissociation of shallow gas hydrates on your model too. Also it would be great to place in the supplementary material a photo of the core showing gas hydrates to confirm their presence and the exact position of the cored interval on the seismic line.

We have added a figure to the supplementary material showing a photo of the core showing gas hydrates to confirm their presence and the exact position of the cored interval on the seismic line.

Indeed, we were not able to recover hydrate from the tongue-shaped gas hydrate stability zone because its top is at 20 m below the seafloor. This is why we sampled at 800 m water depth where the gas hydrates were expected to occur close to the seafloor.

120-126: I checked some of the references used to support the argument that a generalization of the model to the Black sea is justified. Here some checks: Ref 44 gives an approximate constrain on the actual distribution of gas hydrates. I'm wondering why these results have not been taken into account. Ref 47 shows release of gas at an average depth of 850 m suggesting that a stabilization is deeper than what modelled in the present work. They exclude gas hydrate destabilization in the near future. Ref 48 shows GH destabilization and seeps at a depth >900 m (Fig. 8) and far deeper than the modelled GH destabilization zone suggested by the present work. Ref 49 shows triggering of submarine landslides potentially caused by GH destabilization below 1600m, again far below the GH destabilization zone proposed in this work. Ref 52 refers to authigenic carbonates and gas release discovered in a depth range of 400-2000 m. I suggests revisiting the literature carefully and evaluate whether the proposed GH destabilization process is applicable to the entire Black Sea. Based on these references, it seems clear that a speculative extension of the proposed model is not feasible.

We agree with reviewer 3's comments concerning the above 5 references. But these references were used to show that gas hydrates occur in almost the whole Black Sea shallow water depths. These papers cannot use to demonstrate what we consider here because the new important finding of concerning the

degassing associated with the current dissociation of the hydrates between 650 m and 750 m water depth was not known. This is the first time that this specific and non-documented mechanism is described and analyzed. The gas seeps inside the GHSZ shown in refs 47 and 48 and confirmed recently by Riboulot et al. (2017) are not related to hydrate dissociation but to free gas circulation through fractures and faults crossing the GHSZ.

Also, it would be interesting to evaluate whether the proposed GH destabilization process is more impactful than other documented gas expulsion phenomena deriving from other processes.

For the considered 5 kyrs calculation period, we considered the impact of an ongoing proven process related to salt diffusion and we considered unchanged bathymetry and constant temperature and pressure conditions (Past Interglacials Working Group of PAGES, 2016). It was possible to consider in the calculation a seabed temperature or a sea-level change but how to evaluate this T-P evolution over a 5 kyrs calculation period?

Reviewers' comments:

Reviewer #3 (Remarks to the Author):

The paper claims that diffusion of salt in pore space in the first meters of sediments and triggered by changes in salinity of the seawater can destabilise gas hydrates with a subsequent release of methane into the atmosphere.

The idea is novel and of great interest however, as already explain in the previous occasions, the application of the model to the entire black sea is still not convincing. I suggests to review the amount of gas release from the gas hydrate destabilisation zone once a clear distribution of gas hydrates is taken into account.

I prepared a document (attached) with few detailed comments based on the latest version of the manuscript.

Manuscript#: NCOMMS-16-26092C

Corresponding Author: Vincent Riboulot

Title: From a freshwater lake to a salt-water sea causing widespread hydrate dissociation in the Black Sea

Detailed comments

Previous reviewer comment/answer/reviewer comment	(1) Lines 32-35: the BSR, as discussed by Xu and Ruppel 1999, does not coincide to neither the base of the gas hydrate stability zone (GHSZ) nor the top of the free gas zone (FGZ). I suggest referring to this paper for a better significance of the BSR. I would explain also what a negative polarity is. It is negative with respect to what. If the polarity of the seismic data is European the BSR is positive. The BSR, as discussed by Xu and Ruppel 1999 differs from the BSR observed in our study area because velocity analysis of the seismic data shows a sharp decrease in the velocity right under the BSR revealing the presence of free gas trapped under the BSR. Conversely an increase in the velocity right above the BSR suggests the occurrence of hydrates at the base of the GHSZ corresponding to the location of the BSR. The inverse polarity of the BSR is with respect to the polarity of the seafloor. So if the seafloor is positive, the BSR is negative. The polarity of the BSR is negative only if the polarity of the dataset is American (where a decrease of acoustic impedance with depth is represented by a negative reflection coefficient); if the polarity of the dataset is European (where, instead, a decrease of acoustic impedance with depth is represented by a positive reflection coefficient) the BSR is positive. When you say “The BSR represents the base of the Gas Hydrate Stability Zone (GHSZ) that appears as strong, negative-polarity,...” is a valid statement, but you have to define the polarity of the dataset first.
Answer – present version	This is correct, we use the American convention (a decrease of impedance is represented by a negative reflection coefficient). This is now mentioned in the caption of figure 2.

Previous reviewer comment/answer/reviewer comment	(2) Lines 36-38: provide a reliable distribution map of the BSR, which you will use then to infer the distribution of the GH. Without the distribution of the GH in the Black Sea, most of your argument from line 112 onwards are hard to argue (even for a first order evaluation). Reference 1 (fig. 4) shows a map extracted from a work of Vassilev et al 2006 showing evidence of extremely variable GH distribution in the Black Sea; References 26 and 27 show the distribution of GH only in NW Black Sea (both works are based on 2D seismic data only). Reference 26 (fig.1) shows indeed just some areas covered by BSR. The Reference 1 (Meray and Sinayuc, 2016) uses a map from Vassilev et al. (2006) showing in blue the extension of the first BSR covering almost
---	---

	the whole Black Sea; the other colours represent the occurrence of multiple BSRs. In the study of Merey and Sinayuc (2016), the map presented in Fig. 6 shows the volume of CH₄ in hydrates in standard conditions for the Black Sea suggesting the occurrence of hydrates in our region of interest where hydrate may dissociate by salt diffusion. The Reference 26 (Popescu et al., 2006) shows areas covered by BSR in NW Black Sea where 2D seismic data were available. The extended data acquired in the GHASS project in 2015, some of which are shown in our study, show that the BSR is observed in all seismic data located to the north of the Danube canyon at a water depth in agreement with the base of the gas hydrate stability zone. The cartography of this BSR is not the aim of the present paper but it may be added to the supplementary data if you consider it as essential. This is the distribution of BSR on the Black Sea from Vassilev, A., (2006). This distribution map is based on a work of Popescu et al., 2006, which include only 5 areas and 11 seismic segments (thick black lines). The other maps showed in the same work are based on estimated optimistic and pessimistic scenarios and not on observed BSR. Merey and Sinayuc, 2016 used estimated values. It is also worth noting that previously published maps based on factual observations of BSR in the Black Sea have not taken into account, e.g.: Poort et al., 2005; Vassilev & Dimitrov, 2003; Merey et al, 2016. Again, the authors are invited to reconsider the risk of extending the application of their model for the destabilization of the GHSZ to the entire Black Sea.
Answer – present version	In the previous version of the paper we calculated the upper bound of the area where the gas hydrate may decompose due to salinization process. In this upper-bound calculation we assumed that gas hydrate is occurring in the whole Black Sea wherever the thermodynamic conditions are valid. We agree with the reviewer 3 about the possibility to improve the previous extrapolation. In the present version of the paper, the new calculation considers only areas where free gas/gas hydrates have been recovered or inferred (see Supplementary Fig. 2 and references in caption). We have thus modified our estimation of the amount of gas generated by gas hydrate dissociation due to salinization of sediment. Figure 1 and the Supplementary Figure 2 have been modified accordingly and details supporting our working hypothesis are given in the supplementary material file (L12-20 of the Riboulot_marked_SupplementaryInformation file).

Previous reviewer comment/answer/reviewer comment	(3) Line 39: Reference 28 shows evidence of freshening consistent with a dehydration caused by smectite to illite process. Indeed the authors of this paper show that the source of fluid freshening in the central part of the Ulleung Basin (Japan Sea or East Sea) is due to dehydration caused by smectite to illite process. This process was not described in the Black Sea.
--	---

	The dehydration from smectite to illite produce fresh water and not chlorine rich water. Fresh water do not destabilizes GH. The Ulleung Basin cannot be used as analogue.
Answer – present version	Yes we agree, the Ulleung basin is not a good example. We have now removed all the references to this area from the manuscript.

Previous reviewer comment/answer/reviewer comment	(4) Also, it would be interesting to evaluate whether the proposed GH destabilization process is more impactful than other documented gas expulsion phenomena deriving from other processes. For the considered 5 kyrs calculation period, we considered the impact of an ongoing proven process related to salt diffusion and we considered unchanged bathymetry and constant temperature and pressure conditions (Past Interglacials Working Group of PAGEs, 2016). It was possible to consider in the calculation a seabed temperature or a sea-level change but how to evaluate this T-P evolution over a 5 kyrs calculation period? Apologies if I was not clear. What I was trying to ask was related to the effective amount of methane released by the salt diffusion model with respect to other processes of methane release in the Black Sea. For instance, the author show release of methane from a region where the GHSZ is expected to be stable (e.g. Figure 4) and suggest that some amount of gas could have been released from faults (Supplementary Figure 7). So, what is the impact of such processes with respect to the salt diffusion model proposed?
Answer – present version	Our work has allowed to determine a rough indication of the methane released by salt diffusion. Without an important plan for monitoring free gas fluxes at the seafloor level inside but also and especially outside the GHSZ it will be very difficult to evaluate the ratio between methane released by the salt diffusion with respect to other processes. This question could be the starting point of an ambitious future research project in the Black Sea. On the other hand, the impact of the local fault system as a preferential path for gas migration is expected to be negligible because it is an isolated structure and almost unique in our study zone (Riboulot et al., 2017).

REVIEWERS' COMMENTS:

Reviewer #3 (Remarks to the Author):

Reviewer #3

The paper claims that diffusion of salt in pore space in the first meters of sediments and triggered by changes in salinity of the seawater can destabilise gas hydrates with a subsequent release of methane into the atmosphere.

The idea is novel and of great interest. I am really satisfied with this revised version of the manuscript which I fully recommend for publication without further modifications.